# A Method of Estimating Time-to-Recovery for a Disease Caused by a Contagious Pathogen Such as SARS-CoV-2 Using a Time Series of Aggregated Case Reports

**DOI:** 10.3390/healthcare11050733

**Published:** 2023-03-02

**Authors:** Dimitrios-Dionysios Koutsouris, Stavros Pitoglou, Athanasios Anastasiou, Yiannis Koumpouros

**Affiliations:** 1Biomedical Engineering Laboratory, National Technical University of Athens, 15780 Athens, Greece; 2Research & Development, Computer Solutions SA, 11527 Athens, Greece; 3Department of Public and Community Health, University of West Attica, 11521 Athens, Greece

**Keywords:** SARS-CoV-2, time-to-recovery, COVID-19

## Abstract

During the outbreak of a disease caused by a pathogen with unknown characteristics, the uncertainty of its progression parameters can be reduced by devising methods that, based on rational assumptions, exploit available information to provide actionable insights. In this study, performed a few (~6) weeks into the outbreak of COVID-19 (caused by SARS-CoV-2), one of the most important disease parameters, the average time-to-recovery, was calculated using data publicly available on the internet (daily reported cases of confirmed infections, deaths, and recoveries), and fed into an algorithm that matches confirmed cases with deaths and recoveries. Unmatched cases were adjusted based on the matched cases calculation. The mean time-to-recovery, calculated from all globally reported cases, was found to be 18.01 days (SD 3.31 days) for the matched cases and 18.29 days (SD 2.73 days) taking into consideration the adjusted unmatched cases as well. The proposed method used limited data and provided experimental results in the same region as clinical studies published several months later. This indicates that the proposed method, combined with expert knowledge and informed calculated assumptions, could provide a meaningful calculated average time-to-recovery figure, which can be used as an evidence-based estimation to support containment and mitigation policy decisions, even at the very early stages of an outbreak.

## 1. Introduction

The spread of infectious diseases is admittedly an unpredictable process. On a global scientific scale, we are witnessing the resistance of viruses to antibiotics and the emergence of alarming new viruses, most recently the example of SARS-CoV-2, making the stakes of effectively dealing with them higher than ever. Many mathematical methods have been used over time by the global community to predict and measure the multiple factors involved in infectious diseases. In particular, a plethora of modeling techniques have been developed that aim to compensate for the imperfect knowledge collected from large populations and under difficult conditions during the transmission of an infectious disease. Heesterbeek et al. [1], in their extensive study, examined the development of mathematical models used in epidemiology and their use in developing successful public health control strategies. As highlighted in the conclusions of their study, mathematical modeling is the most effective tool in the hands of the scientific community to explore the seemingly intractable complexity of infectious disease dynamics [1].

Despite the remarkable progress that has been made in the areas of prevention and control, infectious diseases continue to pose a global threat to human and animal health, as witnessed by the uncontrolled transmission of COVID-19 and its devastating effects. As is well known in the scientific community, the ecological and evolutionary dynamics of pathogens evolve over a wide range of interconnected temporal, organizational, and spatial scales ranging from hours to months or even years, from cells to entire ecosystems and from local to global. Pathogenic microorganisms have many different modes of transmission, either between individuals of a single species, between multiple hosts, via arthropod vectors, or by persistence in environmental reservoirs. There are many factors that influence transmission and pose global challenges for the prevention and control of these diseases, such as increasing antimicrobial resistance, human connectivity, population growth, urbanisation, environmental change and mismanagement of land, and changing human behaviour. Mathematical models have the potential to address the aforementioned complexity and provide valuable tools for understanding epidemiological patterns and developing and evaluating factors that influence decision-making in global health [1].

Global biosafety depends on the ability of the scientific community to effectively address emerging infectious diseases and threats. In evaluating and responding to the advent of pathogens, mathematical models that incorporate scientific knowledge regarding disease processes will continue to be crucial. Although these methods have numerous limitations and pitfalls, they provide vital information that will contribute to an effective response by the global health community. As a conclusion, mathematical models, through the exploitation of new data streams, can reveal transmission mechanisms and suggest new approaches for the prevention and control of infectious diseases, always in conjunction with the ongoing dialogue between decision-makers and the infectious disease multidisciplinary community, thus contributing to the formulation of national and international public health policy [1].

The current outbreak of severe acute respiratory syndrome-coronavirus 2 (SARS-CoV-2) has set the scientific community an extremely challenging task and provided a rapid lesson in global epidemiology, starting with lessons in case detection and exponential growth globally. This new infectious disease also reminded scientists of the challenges of effective communication as it evolved in a context of complete uncertainty. Although the current pandemic is a rare and unique phenomenon, the virus is observed to exhibit similar behavioral patterns to other viruses. Particular attention should be paid to the fact that the mathematical modelling that has been applied to previous historical influenza pandemics demonstrates that comparing the effects of interventions in different populations is not an effective method of responding to a pandemic. For example, a rapid reduction in coronavirus disease cases in 2019 (COVID-19) can be taken as evidence that interventions were highly effective or even that herd immunity was achieved. Simple mathematical models, however, demonstrate that when there is seasonal change in virus susceptibility or transmission, especially when there is population migration, epidemic dynamics become profoundly opaque. In particular, the pattern of seasonal change is questionable for SARS-CoV-2, for example. Simple linear correlations may result in incorrect inferences and misinterpretations regarding the interventions that are deemed successful and most effective. It is crucial to stress how geographical variation can be seen in the number, timing, and intensity of a pandemic’s waves based on empirical evidence from previous influenza pandemics. To avoid making direct comparisons between the SARS-CoV-2 pandemic and earlier pandemics, mathematical methods are crucial. These changes are caused by differences between populations in their pre-existing immunity and seasonal factors [2].

Going deeper into understanding the transmission of an infectious disease, by ‘pathogen emergence’ scientists are referring to the entire pathway of the virus, from the entry of uniquely identified viruses into the human population to the invasion of established pathogens into new populations and the evolution of drug resistance. In particular, mathematical models relating to emerging pathogens are used to predict the number of outbreaks, investigate transmission mechanisms, and evaluate control options where feasible. As mentioned above there are numerous limitations and weaknesses in the use of these models, often due to lack of data [3]. Indeed, there are many “known unknowns” in the progression patterns of an infectious disease caused by a pathogen that has not been widely spread before and whose characteristics have not been extensively studied. One of them is the time-to-recovery (or recovery time), $t_{rec}$, defined as the time that passes from the onset of the disease before an infected individual moves to the recovered class having fought off the infection. 

An accurate estimation of this time interval is essential when a disease has pandemic characteristics. From a disease modeling perspective, it translates to the probability of an individual moving from the Infected to Recovered class, being dependent on how long they have been infected. It is an essential parameter to any mathematical disease model that contains I→R  dynamics (S→I→R, S→E→I→R, etc.) [4]. Furthermore, it provides a measure of how long any infected individual is likely to transmit the disease, so it could help responding authorities make informed and evidence-based decisions on containment measures/policies, such as quarantine periods, population mobility restrictions, etc.

One of the main restrictions is that, during the outbreak, and due to the extensively multi-centered data flow (from hundreds of distressed healthcare facilities), data about the confirmation, deaths, and recoveries are reported in an aggregated fashion (“Aggregated routine surveillance” [5], i.e., daily totals of cases per category), rendering effectively impossible the data analysis on a per case basis. Furthermore, during the initial period of the outbreak of a previously un-studied pathogen, although there is an urgent need for informed policy decisions, there are not enough data and time to design, perform, and draw conclusions from reliable clinical trials.

In this paper, an algorithmic approach/method is proposed that overcomes the above restrictions, exploring the hypothesis that a reliable and actionable estimate of the time-to-recovery for an infectious disease could be calculated from available reported data, even during the early stages of the outbreak. 

In order to come up with a meaningful approach given such limited data and severe information loss due to their aggregated form, a series of assumptions must be made:The disease progression follows the simple model depicted in Figure 1;It is taken as granted that each case reported is subject to regular follow-up, ensuring that the time of recovery or death will be recorded and appended to the available aggregated cumulative datasets without exception;The mean time to recovery is calculated irrespective of specific patient strata (e.g., age, gender, etc.). This is consistent with the simplification commonly used by disease modelers that the recovery rate (which is the inverse of the infectious period) is constant [4];The mean time to recovery for patients that recover is equal to the average time to death in the mortal case occasions (${\bar{t}}_{rec}={\bar{t}}_{d}$), as it is often considered plausible to assume that mortality occurs towards the end of the infectious period [4]; Confirmed cases that are not yet matched (considered still ill) at the time of analysis will have a time-to-recovery equal to the mean calculated from the matched cases. 

## 2. Materials and Methods

### 2.1. Data Source

In response to the public health emergency caused by SARS-CoV-2, Johns Hopkins University developed an interactive web-based dashboard hosted by their Center for Systems Science and Engineering (CSSE) to visualize and track reported cases in real time [6]. The data sources include the World Health Organization (WHO) [7], the Centers for Disease Control and Prevention (CDC) [8], the European Centre for Disease Prevention and Control (ECDC) [9], and China’s National Health Commission (NHC) [10]. All the data collected and displayed are made freely available on the GitHub repository [11]. The raw data sources, in the form of “comma-delimited/separated files” (CSV) used for the subsequent experiments, are presented in Table 1.

### 2.2. Data Processing

The data preparation part of the algorithm processes the daily cumulative data from the source mentioned above, in order to create a time series with the daily newly reported cases per category. Next, using these time series, it creates three (3) sets that contain one data point for each reported case (confirmed, death, and recovered, respectively), using as a value the index of the date that it was reported. Specifically, in our experimental case, the available data start on 22 January 2020; thus, all cases reported on this date are appended to the set with the value ‘1’, cases reported on 23 January 2020 with the value ‘2’, etc. 

Each confirmed case is matched with the earliest unmatched reported case of death or recovery, whichever comes earlier. The death cases take precedence when there are both unmatched death and recovery cases on the same date. The difference between each pair’s dates is calculated, and the cases that take part in this calculation are removed from the respective sets. This procedure is iterated until there are no death and recovery cases left. At this point, the mean of the intervals of the matched case pairs is calculated.

For the rest of the confirmed cases, the difference from the last available date is calculated. If it exceeds the previously computed average, it is appended to the final set unchanged. If not, it is replaced by the average. 

The final set includes estimated time intervals between time of confirmation and recovery time for all confirmed cases to date. From this complete set, the overall mean time-to-recovery is calculated. As new data reports come in daily, this figure can be updated to provide enhanced estimation accuracy based on a growing body of evidence.

### 2.3. Experiments

To check the main hypothesis experimentally, a computer program was written in Python programming language to apply the algorithm described above to the available data (from 22 January to 9 March 2020). Moreover, there was the ability to choose any particular country, cluster of countries, or global figures to run the calculations. 

The reported results include (i) global figures; (ii) Mainland China as the “cradle” of the infection and the most extensive available dataset; (iii) Italy and Iran as two countries with wide spread of the disease; and (iv) US, UK, Spain, France, Germany, and the Netherlands as countries with highly probable domestic sustained spread of the disease and relevant high quality of healthcare service infrastructure.

## 3. Results

The results of the experimental runs for the selected countries/clusters/regions are presented in Table 2. Furthermore, reported cases diagrams and calculated interval distributions are shown in Figure 2.

The mean time-to-recovery calculated from all globally reported cases is 18.01 days (SD 3.31 days) for the matched cases. Taking under consideration the adjusted unmatched cases as well, based on the assumptions mentioned above, the mean recovery time is adjusted to 18.29 days (SD 2.73 days).

As expected, Mainland China accounts for most reported cases (80,735 out of 113,583, 71.08%) and is the dominant subset, substantially affecting the global figures.

## 4. Discussion

The accuracy of the proposed methodology’s figures is directly dependent on the plausibility of the underlying assumptions, as reported in the Introduction section. Additionally, the specific conditions of this study’s experimental leg, the specific disease (COVID-19), its stage, and the available data, introduce several limitations regarding the approach, methodology, and experiments conducted in this study. As far as the data quality is concerned, it should be noted that they are provided to the public strictly for educational and academic research purposes, as they rely upon publicly available data from multiple sources that do not always agree. Regarding the purpose of this article, the accuracy of the reporting is presumed adequate. However, even if the raw reports are aggregated, confirmed, and appended to the source data files in an accurate and timely manner, there is a significant uncertainty factor due to the under-ascertainment (mild, pauci-symptomatic, and subclinical cases) [12,13,14], and, more importantly, there is no guarantee for the quality or the homogeneity of the methodology used in the originating health facilities that compile and report the case numbers to the surveillance authorities.

This fact, combined with the early stage of the experiments leading to small peripheral datasets, provides a plausible explanation for the variability observed between individual countries. Factors such as delays in first reports due to low initial awareness could explain lower means in some countries at this early stage. However, the cumulative deviation caused by temporary reasons or measurement noise is expected to become smaller with time, and figures will tend to converge. This tendency can be disrupted by systematic affecting factors, such as different definitions of recovery (due to medical approaches, or in some cases, political reasons [15,16]).

Finally, COVID-19 is the first epidemic for which official data of recovered cases are collected and made publicly available, which makes the proposed calculation approach possible, but also means that there is no practical way to evaluate it by applying it to historical data of analogous epidemics, such as SARS and MERS. Thus, the only evaluation method lies in future work involving retrospective case studies with non-aggregated datasets.

Regardless of the limitations mentioned above, a comparison of the estimates derived by the method used in this paper with analogous measurements based on clinical trials and analysis of data collected for more than one year in the pandemic (in contrast with the six weeks used here), demonstrates remarkable accuracy. This could be an indication of the real-world usefulness of the proposed method.

More specifically, Khalili et al. [17] conducted an extensive meta-analysis of 43 studies showing that the estimated mean number of days from the onset of symptoms to recovery was reported in seven studies and the resulting pooled mean was 18.55. Furthermore, Qifang Bi et al. [18] included only confirmed cases identified by the Shenzhen CDC in their research, between 14 January and 12 February 2020, and close contacts of cases confirmed before 9 February 2020. This research team included 391 cases that were older than the general population (mean age 45 years) and found that three cases had died and 225 had recovered (median time to recovery 21 days; 95% CI 20–22) as of 22 February 2020. 

Another related study was carried out by the research group of Barman et al. [19]. They focused on a random sample of 221 COVID-19-positive individuals in India from 1 March 2020 to 25 April 2020 and noted that the average recovery time of COVID-19 patients in India was 25 days (95% C.I. 16 days to 34 days). Moreover, Zhou et al. [20], conducted a retrospective cohort study in which they included 191 patients with laboratory-confirmed COVID-19 from Jinyintan Hospital and Wuhan Pulmonary Hospital (Wuhan, China) who had been discharged or had died by 31 January 2020.

In addition, Tolossa et al. [21] presented a hospital-based retrospective cohort study including 263 adult patients admitted with COVID-19 in Wollega University Referral Hospital from 29 March 2020 through 30 September 2020. This research team concluded that the median recovery time of patients with COVID-19 cases was 18 days, and factors such as older age group, presence of fever, and comorbidity was an independent predictor of delayed recovery from COVID-19. Then, SeyedAlinaghi et al. [22] enrolled 478 patients of a university hospital in Tehran, and the median time to recovery was 14.8 days, noting that in the bivariate analysis, multiple factors, including hypertension, diabetes mellitus, gender, and admission location, significantly contributed to prolonging the recovery period; in multivariate analysis, only dyspnea affected recovery period.

Table 3 shows our estimations side-by-side with measurements from quality literature. Wherever the results were reported as medians with interquartile ranges, they were converted in means and 95% confidence intervals via tools [23] based on [24].

## 5. Conclusions

In this paper, plausible assumptions led to a method that utilizes raw aggregated data, available via surveillance reporting routes even at the early stages of a disease outbreak, in order to calculate, with the minimum possible uncertainty, the average time-to-recovery of an infected individual. This information can be used by health authorities to make informed decisions concerning disease containment measures (quarantine, isolation) at the earliest stages of an outbreak, when it matters the most. The method was experimentally tested, resulting in an estimation of this vital time interval relative to COVID-19, with limited (~6 weeks) data. The preliminary experimental calculations support the hypothesis that the proposed methodology could provide meaningful and actionable estimations to facilitate evidence-based disease spread forecasting and containment measure design. This method can be generalized and applied to any infectious disease outbreak, provided that reporting via aggregated routine surveillance is active.

## Figures and Tables

**Figure 1 healthcare-11-00733-f001:**
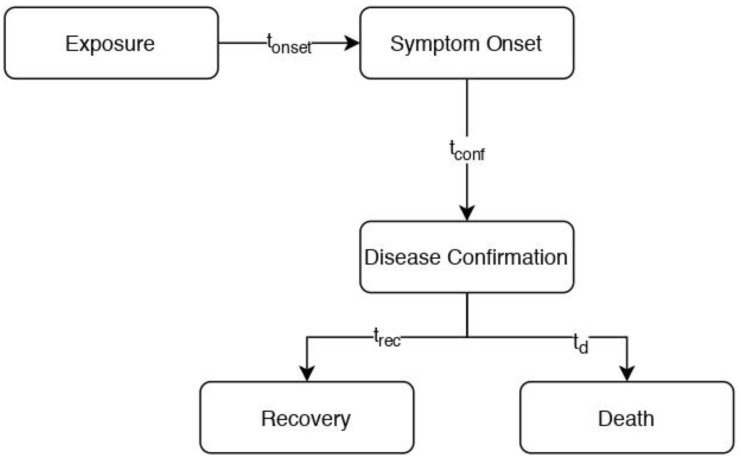
Model of disease progression.

**Figure 2 healthcare-11-00733-f002:**
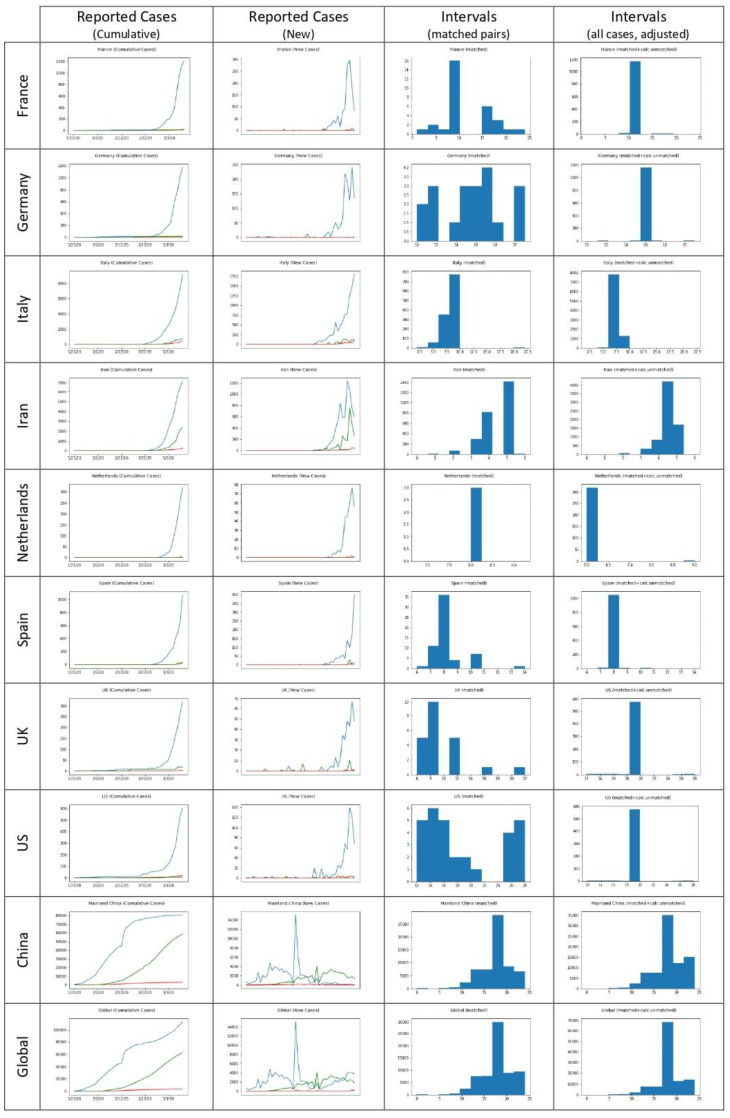
Reported cases, diagrams, and calculated interval distributions.

**Table 1 healthcare-11-00733-t001:** Source data files.

Case Category	File ^a^
Confirmed	time_series_19-covid-Confirmed.csv
Deaths	time_series_19-covid-Deaths.csv
Reported	time_series_19-covid-Recovered.csv

^a^ url path: https://github.com/cssegisanddata/covid-19/tree/master/csse_covid_19_data/csse_covid_19_time_series/{}, accessed on 10 march 2020.

**Table 2 healthcare-11-00733-t002:** Results.

	Matched			All ^a^		
	# ^b^	Mean	SD	# ^b^	Mean	SD
China	61,855	17.81	3.31	80,735	18.52	3.39
Netherlands	3	8	0	321	8.01	0.10
France	31	11.84	5.51	1209	11.84	0.87
US	30	18.77	5.86	605	18.77	1.28
Spain	60	8.18	1.17	1073	8.18	0.27
UK	22	9.59	3.79	321	9.59	0.97
Iran	2631	4.37	0.82	7161	4.40	0.51
Germany	20	15.60	3.30	1176	15.60	0.42
Italy	1188	7.73	1.16	9172	7.74	0.42
Global	66,508	18.01	3.31	113,583	18.29	2.73

^a^ matched plus adjusted unmatched, ^b^ No of cases.

**Table 3 healthcare-11-00733-t003:** Estimations side-by-side with measurements from quality literature.

**Study**	$t_{rec}$ **in Days (95% CI)**	** 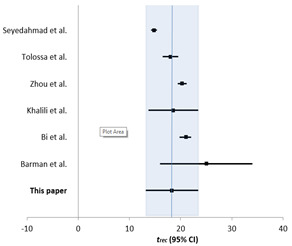 **
Khalili et al. [17]	18.55 (13.69–23.41)	
Bi et al. [18]	21 (20–22)	
Barman et al. [19]	25 (16–34)	
Zhou et al. [20]	20.3 (19.4–21.2)	
Tolossa et al. [21]	18 (16.47–19.52)	
SeyedAlinaghi et al. [22]	14.8 (14.2–15.4)	
This paper	18.29 (13.16–23.41)	

## Data Availability

The code used for the analysis has been made publicly available as Open-Source Software in a GitHub repository: S. Pitoglou, “spitoglou/covid-19-time-to-recovery: Review.” 22 December 2022, doi: 10.5281/ZENODO.7473221. The data used in this research are publicly available by the Center for Systems Science and Engineering (CSSE) of John Hopkins University (GitHub repository: https://github.com/CSSEGISandData/COVID-19 (accessed on 20 February 2023)).

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
