# Peer review of "A Method of Estimating Time-to-Recovery for a Disease Caused by a Contagious Pathogen Such as SARS-CoV-2 Using a Time Series of Aggregated Case Reports"

_healthcare, 2023, doi:10.3390/healthcare11050733_

Round 1

Reviewer 1 Report

It is a very interesting study. However, I have some suggestions worth considering in order to provide better scientific value for the article. First of all I am not sure whether this manuscript should be published as Article - maybe Brief Report or Communication.

Results

Table 2.

I believe presenting the results as medians and interquartile ranges would be more accurate. Mean values are very sensible for small and high values in the results, therefore median i best suited for this kind of data.

What does # mean? I suggest providing a legend to every Table 2

Discussion

The discussion is mainly focused on the study limitations.I agree they should be stated in this section. However, I would suggest developing this section with discussing your results with results obtained by other scientists.

Author Response

We are thankful for the reviewer's feedback and comments. Please find below our point-to-point responses:

Point 1: [Results: Table 2] "I believe presenting the results as medians and interquartile ranges would be more accurate. Mean values are very sensible for small and high values in the results, therefore median i best suited for this kind of data."

Response 1: The use of mean values in Table 2 is consistent with the primary objective of the calculations, i.e., the mean time-to-recovery. Arguably, the use of the median approach is generally more appropriate when the data are not normally distributed. However, in the context of the methodology used in this study (especially given the assumptions for the handling of the unmatched cases), the use of the mean (effectively letting possible "outliers" affect the calculation) was considered a preferable approach to provide an actionable "ballpark" estimation. In the relevant literature, one can find both approaches represented.

Point 2: [Results: Table 2] "What does # mean? I suggest providing a legend to every Table 2"

Response 2: In Table 2, columns with "#" contain the number of respective cases. Relevant reference was added to the table's legend.

Point 3: [Discussion] "The discussion is mainly focused on the study limitations. I agree they should be stated in this section. However, I would suggest developing this section with discussing your results with results obtained by other scientists."

Response 3: The discussion section of the manuscript has been updated to include descriptions of the literature used for the comparisons.

Reviewer 2 Report

The introducion and discussion part could be improved, specially the discussion part. More references are needed throughout the whole manuscript, and the importance and practical applicability of the model reported in the study should be better explained.

The conclusion could also be reformulated in order to highlight the importance of this model if it were to be used an applied by health authorities.

More comments are enclosed in the pdf file.

Author Response

We are thankful for the reviewer's feedback and comments. Please find below our point-to-point responses:

Point 1: "The introduction and discussion part could be improved, specially the discussion part. More references are needed throughout the whole manuscript, and the importance and practical applicability of the model reported in the study should be better explained."

Response 1: The discussion section of the manuscript has been updated to include descriptions of the literature used for the comparisons. More references were added throughout the manuscript, and the practical applicability was highlighted further in the abstract and the "Discussion" and "Conclusion" sections.

Point 2: The conclusion could also be reformulated in order to highlight the importance of this model if it were to be used an applied by health authorities.

Response 2: The "Conclusion" section is updated accordingly.

Point 3: More comments are enclosed in the pdf file.

Response 3:

a) the abstract was updated to provide more clarity,

b) the proposed wording corrections were made,

c) To our knowledge there is no published precedent of analogous calculations. Time-to-recovery is mainly calculated in a clinical study setting. Also, the practice of aggregated reporting has not changed substantially.

d) The main actionable value of the proposed method is providing an early estimate of the time-to-recovery. As such, it is essential to provide a meaningful (close to the actual) measure using limited data (a few weeks into the pandemic). Hence, limited data were used to provide the specific proof-of-concept.

e) indeed, new variants with completely different progression profiles could present different time-to-recovery. However, this is a uniform problem with all calculation efforts, regardless of the methodology used.

f) To our knowledge there is not an efficient way to compensate for the systemic problem of the lack of homogeneity in the reporting, as there is not a consistent pattern to be taken into consideration.

g) The discussion section of the manuscript has been updated to include descriptions of the literature used for the comparisons. 

Round 2

Reviewer 2 Report

Thank you for providing the reviewed version and for doing the suggested changes accordingly, as well as explaining the questions raised by me.

I believe now the manuscript is good for publication.